# Missed Opportunities: A Retrospective Study of Hepatitis C Testing in Hospital Inpatients

**DOI:** 10.3390/v16060979

**Published:** 2024-06-18

**Authors:** Christine Roder, Carl Cosgrave, Kathryn Mackie, Bridgette McNamara, Joseph S. Doyle, Amanda J. Wade

**Affiliations:** 1Barwon South West Public Health Unit, Barwon Health, Geelong 3220, Australia; bridgette.mcnamara@barwonhealth.org.au; 2Centre for Innovation in Infectious Disease and Immunology Research (CIIDIR), Institute for Mental and Physical Health and Clinical Translation (IMPACT), School of Medicine, Deakin University, Geelong 3220, Australia; 3Gastroenterology Department, Barwon Health, Geelong 3220, Australia; carl.cosgrave@barwonhealth.org.au; 4Pharmacy Department, Barwon Health, Geelong 3220, Australia; kate.mackie@barwonhealth.org.au; 5Pharmacy Department, Alfred Health, Melbourne 3004, Australia; 6Centre for Epidemiology and Biostatistics, Melbourne School of Population and Global Health, University of Melbourne, Melbourne 3010, Australia; 7Disease Elimination Program, Burnet Institute, Melbourne 3004, Australia; joseph.doyle@burnet.edu.au; 8Department of Infectious Disease, Alfred Health, Melbourne 3004, Australia

**Keywords:** hepatitis C, hepatitis C testing, hepatitis C care, retention in care, cascade of care, people who use drugs, hospitals, micro-elimination

## Abstract

Increasing testing is key to achieving hepatitis C elimination. This retrospective study aimed to assess the testing cascade of patients at a regional hospital in Victoria, Australia, who inject drugs or are living with hepatitis C, to identify missed opportunities for hepatitis C care. Adult hospital inpatients and emergency department (ED) attendees from 2018 to 2021 with indications for intravenous drug use (IDU) or hepatitis C on their discharge or ED summary were included. Data sources: hospital admissions, pathology, hospital pharmacy, and outpatients. We assessed progression through the testing cascade and performed logistic regression analysis for predictors of hepatitis C care, including testing and treatment. Of 79,923 adults admitted, 1345 (1.7%) had IDU-coded separations and 628 (0.8%) had hepatitis C-coded separations (N = 1892). Hepatitis C virus (HCV) status at the end of the study was unknown for 1569 (82.9%). ED admissions were associated with increased odds of not providing hepatitis C care (odds ratio 3.29, 95% confidence interval 2.42–4.48). More than 2% of inpatients at our hospital have an indication for testing, however, most are not being tested despite their hospital contact. As we work toward HCV elimination in our region, we need to incorporate testing and linkage strategies within hospital departments with a higher prevalence of people at risk of infection.

## 1. Introduction

Globally, 57.8 million people live with hepatitis C and are at risk of cirrhosis and liver cancer [1]. In 2016, the World Health Organization (WHO) called for the elimination of hepatitis C as a public health threat by 2030 through harm reduction strategies and direct-acting antiviral treatments (DAAs). Australia is a global leader in progress toward hepatitis C elimination due to unrestricted access to DAAs subsidized by the government, with universal prescribing under the national government-funded Pharmaceutical Benefits Scheme [2,3,4]. This extends access to hepatitis C treatment to include general practice, community health services, drug and alcohol services, needle and syringe programs, and mental health services [5,6,7,8]. Initial uptake of DAAs in Australia was sizeable, with 32,503 people treated in 2016; however, treatment uptake has steadily declined to just 6474 people in 2021 [9]. Furthermore, at the end of 2022, of an estimated 74,400 people living with hepatitis C in Australia, 81% were diagnosed, and of those diagnosed, 75% (45,180 people) had also had an RNA test—demonstrating the gap in the hepatitis C care cascade and the need to engage and retain people in the testing continuum [10].

Innovative methods to increase hepatitis C testing and linkage to care and treatment are required for individuals to realize the benefits of treatment, reduce onward transmission, and achieve the WHO elimination targets in Australia [11,12,13]. The Barwon South West region (BSW) of Victoria, part of the Western Victoria Primary Health Network (PHN), has the highest rate of hepatitis C treatment uptake in Australia (65.38%) and was the only PHN to achieve the 2022 National Strategy treatment target [14]. The BSW micro-elimination program has supported community-based hepatitis C testing and treatment via regional outreach nursing support, remote general practitioner consultation pathways, testing, and linkage to care at a needle and syringe program (NSP), and, recently, an enhanced notification system within the Local Public Health Unit (LPHU) [5,6,15].

Micro-elimination of hepatitis C involves tailoring health resources to meet the needs of a population group (for example, people in a geographic area or people who inject drugs (PWIDs)) to achieve goals that contribute collectively to the national elimination goals. Using local health knowledge, targeted testing and treatment strategies can be developed that will overcome barriers to care experienced by the local population. Understanding barriers and gaps in health service coverage is essential to develop targeted testing and treatment strategies [12,16]. There is keen interest in the role emergency department (ED) and hospital hepatitis C opt-out or risk-based testing and linkage to care programs play in achieving elimination [17]. The effects of such interventions are variable, mirroring the geographic variability in the epidemic and healthcare environment [18,19]. In Australia, despite universal access to DAAs, to date there are regulatory barriers that inhibit prescribing treatment for hospital inpatients. This underlines the need for a micro-elimination approach [12,16]. Our study aimed to assess the proportion of inpatients who inject drugs or are living with hepatitis C that completed testing and engaged in care facilitated by the admission in BSW, a region with high treatment uptake.

## 2. Materials and Methods

### 2.1. Study Setting and Population

University Hospital Geelong (UHG) is a regional, publicly funded, tertiary hospital in BSW servicing a population of 500,000 people [20]. Persons born in non-English-speaking countries make up 8.6% of the population, and Aboriginal and Torres Strait Islander people (henceforth Aboriginal) make up 1.4% of the population [21]. Estimated prevalence of hepatitis C infection in BSW ranges from 0.74% to 0.34% [14]. In BSW, hepatitis C testing is performed via venipuncture for hepatitis C virus (HCV) antibody and, if antibody positive, a second venipuncture for an HCV RNA test [22]. Reflexive testing is limited, but clinical trials for point-of-care testing are active in the region [23]. Treatment is accessible through general practitioners (GPs), specialist services, the viral hepatitis outreach nurse, and some community-based drug and alcohol services and NSPs. Support is available to GPs via the viral hepatitis outreach nurse and through the HealthPathways electronic platform, hosted by the PHN [5,6].

### 2.2. Study Design

All adults admitted to UHG and ED attendees (known collectively as inpatients henceforth) between November 2018 and November 2021, at risk of hepatitis C due to a history of injecting drug use (IDU) or with a history of hepatitis C, were included in the study. An inpatient admission or ED attendance was defined as an episode. Episodes were categorized by separation coding into two groups—persons with a history of injecting drug use (IDU)-related and hepatitis-C-related. Separation coding was derived from the inpatient’s hospital discharge or ED summary, which includes medical notes, investigations, procedures, and diagnoses made during the episode. Separation codes are applied according to the International Classification of Diseases, Tenth Revision (ICD-10) [24] (see Appendix A). As there is no ICD-10 code for IDU, a list of codes that have been validated as indicators for IDU were used [25]. The codes for the group with a history of hepatitis C included patients who reported a history of hepatitis C, patients with laboratory evidence of hepatitis C antibody detected but no RNA, patients with laboratory evidence of chronic hepatitis infection, and a variety of other test combinations, as discussed below.

A retrospective cohort study of hepatitis C care by episode and inpatient was undertaken. Data arising during the study period, pre- or post-admission, for study inpatients were obtained from the hospital pharmacy, outpatient specialist clinic, and pathology. Data included hepatitis C virus (HCV) antibody and RNA tests ordered by UHG staff, DAA prescriptions written at the specialist outpatient clinic or outreach nurse-led care, and DAA prescriptions dispensed by the UHG pharmacy.

Receipt of hepatitis C care was defined as documentation of HCV antibody or RNA test, provision of a script for DAAs, or dispensing of DAAs from the UHG pharmacy. Hepatitis C care was further defined as measured, historical, or inferred. Care was measured if it occurred during or following a patient’s first episode during the study period and historical if it occurred prior to their first episode. Care was inferred if a subsequent step in the hepatitis C care cascade occurred during the study period; i.e., an RNA test ordered during the study period infers the patient is HCV antibody positive, as an RNA test is funded via the Medicare Benefits Schedule only if an individual is HCV antibody positive, and DAA treatment is funded via the Pharmaceutical Benefits Schedule only if an individual has chronic hepatitis C. Hepatitis C exposure was defined as a positive HCV antibody test or hepatitis C separation coding. Hepatitis C was defined as a positive HCV RNA test. Hepatitis C status was classified as HCV RNA positive, HCV RNA negative or HCV antibody negative, HCV antibody positive and RNA unknown, or unknown. A missed opportunity was defined as an episode in which a study inpatient did not receive hepatitis C care (as defined above).

The primary study outcomes were to determine the proportion of inpatients (1) who inject drugs or have a history of injecting drug use that are engaged in hepatitis C testing and care in association with their hospital admission and (2) predictors of receiving care. Furthermore, we aimed (3) to determine the proportion of inpatients living with hepatitis C who engaged in hepatitis C testing and care in association with their hospital admission.

### 2.3. Data Analysis

Data collected included name, hospital record number, date of birth, sex, admission date, discharge date, ICD-10 code, admitted specialty, pathology test date, pathology test performed (i.e., HCV antibody or HCV PCR), test result, dispensing date, and generic name of drug. Data management and analysis were performed using StataIC17 (College Station, TX, USA). Descriptive analysis included count and summary statistics reported as means, medians, and proportions as appropriate. A univariate logistic regression analysis reported as odds ratios (ORs) and OR adjusted for age and sex (aOR) with 95% confidence intervals (95% CI and a95% CI, respectively) was used to assess relationships between admitted specialty, key populations, length of stay, and hepatitis C care. 

## 3. Results

### 3.1. Inpatient Characteristics

Of 79,923 inpatients, 1892 (2.3%) had at least one relevant episode; 628/1892 (33.2%) had hepatitis C-coded episodes, and 1345/1892 (71.1%) had IDU-related coded episodes. The total number of episodes with IDU-related coding was 1643 (1.2 episodes per patient) with a median length of stay of 2.58 days (range 1–88). Mean age was 40 years (range 18–96), proportion female was 54.2%, and 59 (4.4%) identified as Aboriginal. The total number of episodes with hepatitis C coding was 1214 (1.9 episodes per patient), with a median length of stay of 4.6 days (range 1–124). Mean age was 49 years (range 18–81), proportion female 36.3%, and 54 (8.5%) identified as Aboriginal. Appendix B provides descriptive statistics of patient demographics and episode characteristics by ICD-10 code and admitted unit.

### 3.2. Hepatitis C Testing Cascade

At the end of the study period, the hepatitis C status for inpatients with hepatitis C or IDU-related episodes (*n* = 1892) was: 1569/1892 (82.9%) unknown; 88/1892 (4.7%) HCV antibody positive, RNA unknown; 70/1892 (3.7%) HCV antibody negative; 64/1892 (3.4%) HCV RNA not detected; and 101/1892 (5.3%) HCV RNA detected. A total of 323/1892 (17.1%) had received hepatitis C care.

#### 3.2.1. Inpatients with IDU-Related Episodes

Of the 1345 inpatients with IDU-related episodes, 98 (7.3%) had an antibody test; 87/98 (88.8%) were documented, and 11/98 (11.2%) were inferred; 38 of 98 (38.8%) were antibody positive, and 60/98 (61.2%) were antibody negative. A total of 23/38 (60.5%) inpatients with a positive antibody had an RNA test; 22/23 (95.7%) were documented, and 1/23 (4.3%) was inferred. Finally, 14/23 (60.9%) were RNA detected and 9/23 (39.1%) RNA not detected. 

Missed opportunities for hepatitis C care occurred for 92.3% (1242/1345) of inpatients with IDU-related episodes. The testing cascade for inpatients with IDU-related episodes is shown in Figure 1.

The median length of stay (LOS) for IDU-related episodes was 2.6 days (range 1–88). The median LOS for IDU-related ED episodes was 1 day (range 1–2). The LOS for other specialties is reported in Appendix B.

The emergency department (ED) had the highest number of IDU-related episodes, with 820 episodes for 694 inpatients. Inpatients with mental health episodes (all UHG psychiatric wards) had the highest rate of antibody testing performed (8/91, 8.8%), while hepatitis specialist (infectious disease and gastroenterology) and obstetrics and gynecology had the lowest (0/12 and 0/24, respectively). Antibody positivity rates ranged from 26.9% (ED) to 54.5% (mental health). The cascade of care per specialty for patients with IDU-related episodes is shown in Table 1.

#### 3.2.2. Inpatients with Hepatitis C-Coded Episodes 

Of 628 inpatients with hepatitis C-coded episodes, 239 (38%) had an antibody test; 181/239 (75.7%) were documented and 58/239 (24.3%) were inferred. The results were 229/239 (95.8%) antibody positive and 10/239 (4.1%) antibody negative.

Of 229 inpatients with a positive antibody, 151 (65.9%)had an RNA test: 143/151 (94.7%) were documented and 8/151 (5.3%) were inferred. The results were 90/151 (59.6%) RNA detected and 61/151 (40.4%) RNA not detected.

Of 628 inpatients with hepatitis C-coded separations, 246 (60.8%) had evidence of hepatitis C care measured by a known hepatitis C status at the end of the study period, and 382/628 (60.8%) inpatients had no evidence of hepatitis C care. The testing cascade of inpatients with hepatitis C-coded episodes is shown in Figure 2.

The median LOS for IDU-related episodes was 4.6 days (range 1–124). The median LOS for IDU-related ED episodes was 1 day (range 1–1). The LOS for other specialties is reported in Appendix B.

Wards included under the general medicine specialty (General Medicine) had the highest number of hepatitis C-coded episodes with 248 for 154 inpatients. Surgical (general and specialty surgical wards) had the highest number of inpatients with hepatitis C-coded episodes with 231 in 187 inpatients. Inpatients with general medicine episodes had the highest rate of antibody testing (68/154, 44.2%), while ED had the lowest (13/81, 16%). Obstetrics and gynecology had the highest rate of RNA testing (12/18, 66.7%), while hepatitis specialists had the lowest (21/45, 46.7%). Antibody positivity rates ranged from 82.4% (other specialty medicine) to 96.1% (general medicine), and rates of RNA detected ranged from 56.9% (general medicine) to 76.9% (obstetrics and gynecology). The testing cascade by specialty for inpatients with hepatitis C-coded episodes is shown in Table 2.

### 3.3. Predictors of Hepatitis C Care—Identifying Missed Opportunities

The adjusted and unadjusted odds ratios for hepatitis C care by specialty, LOS, Indigenous status, age, sex, and IDU-related admission are shown in Table 3. Predictors of higher antibody testing rates were mental health unit admission (aOR 2.12, a95% CI 1.24–3.63), identifying as Aboriginal (aOR 1.76, a95% CI1.09–2.84), male sex (OR 1.59, 95% CI 1.17–2.16), older age (for each increase in age by one year, OR increased 1.01, 95% CI 1.00–1.02), and increased length of stay (for each increase in length of stay by one day, aOR increased 1.04, a95% CI1.02–1.06). Predictors of higher RNA testing rates were obstetrics and gynecology unit admission (aOR 4.38, a95% CI 1.55–12.37) and increased length of stay (for each increase in length of stay by one day, aOR increased 1.03, a95% CI 1.01–1.05).

Predictors of hepatitis C care were mental health unit admission (aOR 2.23, a95% CI 1.36–3.66), identifying as Aboriginal (aOR 1.64, a95% CI 1.04–2.57), male sex (OR 1.64, 95% CI 1.24–2.17), older age (for each increase in age by one year, OR increased 1.01, 95% CI 1.00–1.02), and increased length of stay (for each increase in length of stay by one day, aOR increased 1.04, a95% CI 1.03–1.06). Predictors of not receiving hepatitis C care were ED admissions (aOR 3.29, a95% CI 2.42–4.48) and episodes with an ICD-10 code of poisoning by drugs (aOR 6.46, a95% CI 4.79–8.71).

The adjusted and unadjusted odds ratios for HCV antibody testing, HCV RNA testing, and not receiving HCV care are shown in Appendix C.

## 4. Discussion

This study found that more than 2% of inpatients at our hospital had admissions associated with IDU or hepatitis C. However, most are not being tested or linked to care despite their hospital contact. As we work toward HCV elimination in our region, these data underscore the need for a testing and linkage strategy within the hospital, with a focus on departments with a higher prevalence of people at risk of infection who are not routinely engaged in care (such as PWID and homeless people), for example, the ED [17,26].

ED attendees had the highest odds of not receiving hepatitis C care, which is consistent with findings in a metropolitan setting [27]. Given that longer hospital stays increased the odds of receiving hepatitis C care (OR: 10.5, 95% CI 1.03–1.06), the low levels of hepatitis C care in the ED may be due to the short duration of stay as well as competing medical priorities. Bloodborne virus screening in EDs has been shown to be an acceptable approach to increase screening rates [17,28,29,30]. Most studies found there was a considerable increase in HCV antibody or RNA testing, but overall uptake of screening ranged from 24% to > 85% of eligible patients [17,28,29,30,31,32,33]. The prevalence of hepatitis C among ED attendees was higher than that of the general population, and universal screening identifies a significant number of people living with hepatitis C who would have been missed with risk-based screening [17,34]. Universal screening rather than clinical testing was accepted by patients who felt this normalized a culture of testing and reduced the stigma associated with BBV infections [17,28]. However, some health system funding structures present a barrier to universal screening in EDs, and cost effectiveness will depend on local RNA prevalence [31,34]. As an alternative, an Australian study of automated guideline-based screening in EDs yielded a prevalence of 1.0% (51/5000 tests RNA detected), of which 12 were new diagnoses [33].

In addition to increasing hepatitis C testing in EDs, numerous other actions are required to eliminate hepatitis C in hospitals [35]. Linkage to treatment is frequently reported to be one of the most problematic steps in the hepatitis C care cascade, with high rates of loss to follow-up, especially when people are referred to different institutions or providers [36]. Hepatitis C care provided in a ‘one stop shop’ in the community may now have a hospital equivalent. The findings of the OPPORTUNITI-C study demonstrate the statistical superiority of commencing DAA for PWID in hospital when compared with standard referral for outpatient care post discharge [37]. Other successful linkage-to-care studies include dedicated nurses or peer workers for health system navigation, inclusive of harm reduction education [17,38,39].

The adjusted odds ratio for receiving hepatitis care by specialty unit varied. This study and our prior work [27] identified higher rates of antibody testing in mental health units, presumably related to hepatitis C being over-represented in this group and an increased awareness of this in attending staff [7,40]. Mental health services are increasingly being identified as an opportunity to provide hepatitis C care [6,7,41]. A recent study in Australia showed a DAA treatment uptake of 46% in mental health inpatients with hepatitis C [42]. Antenatal inpatients, in whom there is a relatively high rate of hepatitis C testing due to inclusion in standard antenatal serology, would benefit from a collaborative care model that coordinates hepatitis C treatment and care post pregnancy, inclusive of infant testing, similar to those in place for hepatitis B [43].

A key limitation of this study, by design, was limiting it to inpatients with hepatitis C or IDU-related coding and not including community-based testing and treatment. This might underestimate eventual testing or linkage by community or primary care providers. These data are also limited in that medical staff may not routinely ask or document in the medical record if an inpatient has a history of injecting drug use, and if asked, an inpatient may not disclose, especially with a new healthcare provider. In addition, risk factors for hepatitis C acquisition other than injecting drug use were not considered in this study.

## 5. Conclusions

Micro-elimination requires a data-driven approach to develop strategies that target gaps in hepatitis C care. This study identified a scope to improve hepatitis C testing in hospital inpatients, particularly in EDs, working within medical priority and length of stay limitations. Increasing hepatitis C testing in at-risk populations enables those infected to realize the benefits of DAA treatment and contributes to micro-elimination in the region.

## Figures and Tables

**Figure 1 viruses-16-00979-f001:**
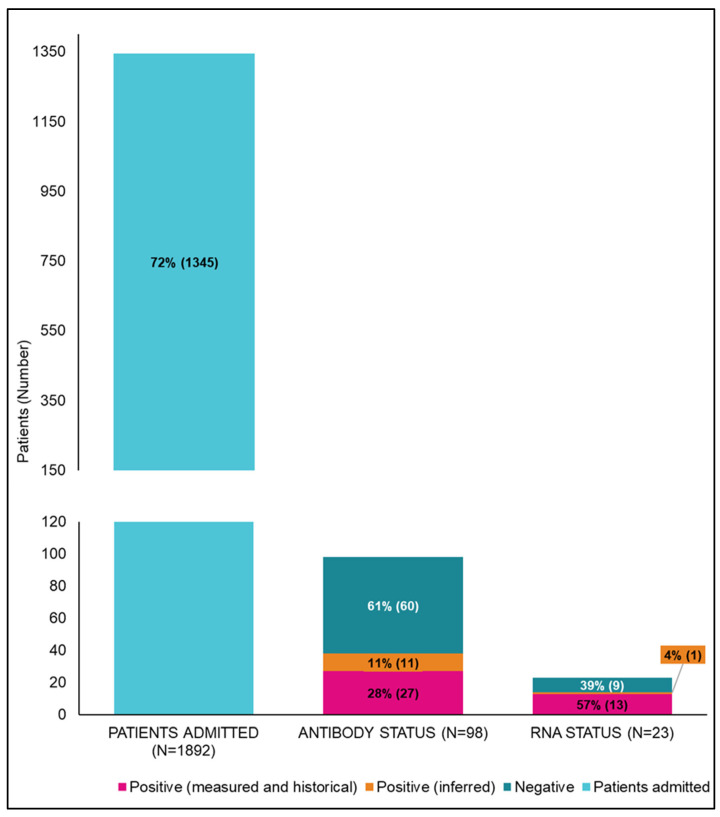
Cascade of care for patients with an indication of injecting-drug-use-related separations at University Hospital Geelong from 2018 to 2021. *n* = 1345 documented (measured and historical) positive test results for hepatitis C antibody, and RNA tests are shown in pink, with inferred positive tests shown in orange.

**Figure 2 viruses-16-00979-f002:**
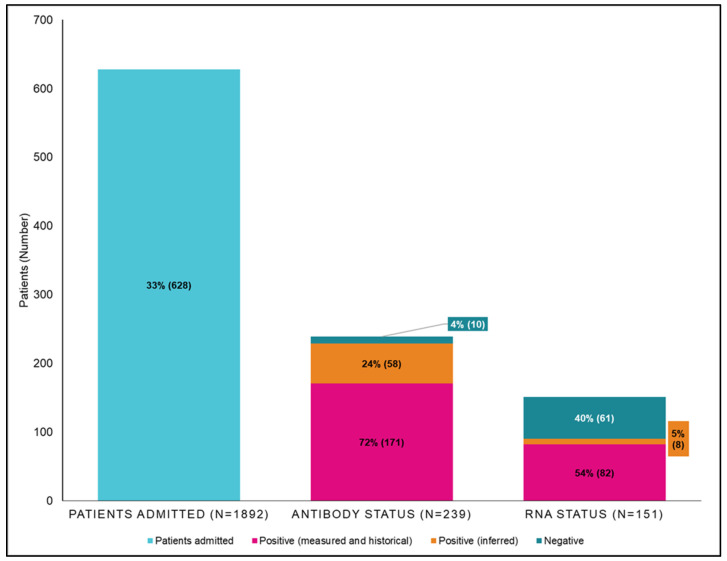
Cascade of care for patients with hepatitis C-coded separations at University Hospital Geelong from 2018 to 2021. *n* = 628 documented (measured and historical) positive test results for hepatitis C antibody, and RNA tests are shown in pink, with inferred positive tests shown in orange.

**Table 1 viruses-16-00979-t001:** The hepatitis C care cascade by specialty for patients with injecting-drug-use-related episodes, 2018–2021.

Specialty	Episodes (*n* = 1643)	Unique Patients (*n* = 1345)	Antibody Testing, *n* (% of Patients)	Antibody Positive, *n* (% of Total Antibody Testing)	RNA Testing, *n* (% of Antibody Positive)	RNA Positive, *n* (% of Total RNA Testing)
Hepatitis specialist	14 (0.9%)	12 (85.7%)	0 (0%; 0)	0 (0%)	0 (0%)	0 (0%)
Emergency	820 (49.9%)	694 (84.6%)	24 (3.5%)	14 (26.9%)	2 (14.3%)	3 (60%)
Surgical	121 (7.4%)	111 (91.7%)	5 (4.5%)	7 (50%)	0 (0%)	2 (50%)
General Medicine	371 (22.6%)	325 (87.6%)	14 (4.3%)	16 (41%)	2 (12.5%)	3 (37.5%)
Mental Health	103 (6.3%)	91 (88.3%)	8 (8.8%)	6 (54.5%)	1 (16.7%)	1 (100%)
Obstetrics and Gynecology	26 (1.6%)	24 (92.3%)	0 (0%)	0 (0%)	0 (0%)	0 (0%)
Other specialty medicine	188 (11.4%)	139 (73.9%)	6 (4.3%)	6 (28.6%)	1 (16.7%)	5 (83.3%)
Total	1643	1345	98 (7.3%)	38 (38.8%)	23 (60.5%)	22 (95.7%)

**Table 2 viruses-16-00979-t002:** The testing hepatitis C care cascade by specialty for patients with hepatitis C-coded episode, 2018–2021.

Specialty	Admissions (*n* = 1214)	Unique Patients (*n* = 628)	Antibody Testing, *n* (% of Patients)	Antibody Positive, *n* (% of Total Antibody Testing)	RNA Testing, *n* (% of Antibody Positive)	RNA Positive, *n* (% of Total RNA Testing)
Hepatitis specialist	188 (15.5%)	108 (57.4%)	27 (25%)	45 (90%)	21 (46.7%)	23 (65.7%)
Emergency	101 (8.3%)	81 (80.2%)	13 (16%)	26 (92.9%)	15 (57.7%)	11 (64.7%)
Surgical	231 (19%)	187 (81%)	37 (19.8%)	52 (89.7%)	30 (57.7%)	27 (67.5%)
General Medicine	248 (20.4%)	154 (62.1%)	68 (44.2%)	98 (96.1%)	55 (56.1%)	37 (56.9%)
Mental Health	77 (6.3%)	60 (77.9%)	21 (35%)	35 (89.7%)	20 (57.1%)	18 (69.2%)
Obstetrics and Gynecology	49 (4%)	34 (69.4%)	7 (20.6%)	18 (90%)	12 (66.7%)	10 (76.9%)
Other specialty medicine	320 (26.4%)	133 (41.6%)	27 (20.3%)	42 (82.4%)	21 (50%)	19 (63.3%)
Total	1214	628	239 (38%)	229 (95.8%)	151 (65.9%)	90 (59.6%)

**Table 3 viruses-16-00979-t003:** Adjusted and unadjusted odds ratios for hepatitis C care by specialty, length of stay, indigenous status, age, sex, and indication of injecting-drug-use-related admission.

	OR	95% CI	aOR	a95% CI
Emergency	**0.12**	**0.06–0.23**	**0.13**	**0.06–0.25**
Gen Med	1.42	0.97–2.07	**1.46**	**1.00–2.13**
Hepatitis Specialist	1.05	0.61–1.82	0.97	0.56–1.69
Mental Health	**2.23**	**1.38–3.60**	**2.23**	**1.36–3.66**
Surgery	0.73	0.44–1.21	0.69	0.42–1.45
O&G	1.83	0.92–3.62	2.43	1.17–5.06
Length of stay	**1.05**	**1.03–1.06**	**1.04**	**1.03–1.06**
Indigenous status	**1.622**	**1.03–2.54**	**1.64**	**1.04–2.57**
Age at admission	**1.01**	**1.00–1.02**		
Sex	**1.64**	**1.24–2.17**		

Odds ratio (OR); odds ratio adjusted for age and sex (aOR); 95% confidence interval (95% CI); adjusted 95% confidence interval (a95% CI). Bold values indicate significance.

## Data Availability

The data used for this study were individual patient data. As such, they have not been made available, to protect the privacy of the patients involved. Aggregated data extracts may be available on request.

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
