# Peer review of "Missed Opportunities: A Retrospective Study of Hepatitis C Testing in Hospital Inpatients"

_viruses, 2024, doi:10.3390/v16060979_

Round 1

Reviewer 1 Report

Comments and Suggestions for Authors

This study aimed to assess the proportion of inpatients at risk of hepatitis C or living with hepatitis C who completed testing and engaged in care facilitated by the admission and to scope the role of inpatient testing at a regional hospital in Victoria (Australia). A total of 79,923 adult inpatients were studied, 1345 (1.7%) were injecting drug users (IDUs) and 628 (0.8%) had hepatitis C. Hepatitis C virus (HCV) status at the end of the study was unknown for 1569 (82.9%). The manuscript contains five keywords, two figures, three tables, three appendix or supplementary tables, and thirty-five references. Overall, it is a correct and well-conducted paper.

General comments
This study highlighted that micro-elimination requires a data-driven approach to develop strategies that target gaps in hepatitis C care. This study identified the scope to improve hepatitis C testing in hospital inpatients, particularly in the Emergency Department, working within medical priority and length of stay limitations. Increasing hepatitis C testing in at risk populations enables those infected to realise the benefits of direct-acting antiviral treatment, and contributes to micro-elimination in the region of Victoria (Australia). The study is methodologically well-established. The data management and statistical analysis are appropriate according to the approach of the study. The results are well presented, being easy to read and interpret them. In the discussion section, the results of this study are adequately contrasted with those obtained by other researchers. A justifying explanation of the results is also provided. The authors revealed the main limitation of this study by design was limited to inpatients with hepatitis C or IDU-related coding and did not include community-based testing and treatment. This might underestimate eventually testing or linkage by community or primary care providers. The manuscript also includes an appropriate concluding section.

Some additional remarks are made on different sections of the manuscript.

Abstract
Please do not place reference numbers "[1,2]" in this section. Move them to the introduction section.
Abbreviations and acronyms, even well-known ones, should be explained the first time they are used, eg. “HCV”.

Keywords
The manuscript presents five keywords. For keywords, where possible, please use Medical Subject Headings terms (MeSH Terms). Strictly, none of them is a MeSH term. Some alternative MeSH terms proposed could be “drug users” better than “people who use drugs”,  “hepatitis C”, or "hepacivirus". Nevertheless, these suggestions about keywords are optional, not mandatory.

Other manuscript sections
Abbreviations and acronyms, even well-known ones, should be explained the first time they are used, eg. “UHG” (page 2, line 82); “LOS” (page 5, line 153).
Page 3, line 133 and following. Please, describe analytical statistics (odds ratios, etc.).
Page 2, line 50. According to the journal’s guidelines, references must be numbered in order of appearance in the text. Reference number 8 should be followed by reference number 9, not number 10 (line 52). Nevertheless, reference number 9 appears below on line 58. Please correct this and place reference number 9 in its proper place after reference number 8.
Page 9, lines 281 to 283. When describing supplementary material, it is coded as Table S1, Table S2, and Table S3 instead of Appendix A (=Table S1), Appendix B (=Table S2), and Appendix C (=Table S3). Please, use only one coding or "Supplementary Table" or "Appendix".

References
Total number of the manuscript references: 35.
This section requires a comprehensive review. The references should be checked carefully to transcribe them accurately. The reference format does not exactly match the journal’s reference format (ACS style guide). The journal's guidelines for reference format recommend using the Abbreviated Journal Name, not the Full Journal Name. The abbreviated journal name is only given in reference number 8; in the rest of the references, the full journal name is given. Please correct this.
According to the journal’s guidelines, references should be described as follows, depending on the type of work:

·        Journal Articles:
1. Author 1, A.B.; Author 2, C.D. Title of the article. Abbreviated Journal Name YearVolume, page range.

·        Books and Book Chapters:
2. Author 1, A.; Author 2, B. Book Title, 3rd ed.; Publisher: Publisher Location, Country, Year; pp. 154–196.
3. Author 1, A.; Author 2, B. Title of the chapter. In Book Title, 2nd ed.; Editor 1, A., Editor 2, B., Eds.; Publisher: Publisher Location, Country, Year; Volume 3, pp. 154–196.

·        Unpublished materials intended for publication:
4. Author 1, A.B.; Author 2, C. Title of Unpublished Work (optional). Correspondence Affiliation, City, State, Country. year, status (manuscript in preparationto be submitted).
5. Author 1, A.B.; Author 2, C. Title of Unpublished Work. Abbreviated Journal Name year, phrase indicating stage of publication (submittedacceptedin press).

·        Unpublished materials not intended for publication:
6. Author 1, A.B. (Affiliation, City, State, Country); Author 2, C. (Affiliation, City, State, Country). Phase describing the material, year. (phase: Personal communication; Private communication; Unpublished work; etc.)

·        Conference Proceedings:
7. Author 1, A.B.; Author 2, C.D.; Author 3, E.F. Title of Presentation. In Title of the Collected Work (if available), Proceedings of the Name of the Conference, Location of Conference, Country, Date of Conference; Editor 1, Editor 2, Eds. (if available); Publisher: City, Country, Year (if available); Abstract Number (optional), Pagination (optional).

·        Thesis:
8. Author 1, A.B. Title of Thesis. Level of Thesis, Degree-Granting University, Location of University, Date of Completion.

·        Websites:
9. Title of Site. Available online: URL (accessed on Day Month Year).
Unlike published works, websites may change over time or disappear, so we encourage you create an archive of the cited website using a service such as WebCite. Archived websites should be cited using the link provided as follows:
10. Title of Site. URL (archived on Day Month Year).

For further information about the reference format proposed by the journal, please, consult the following link: https://www.mdpi.com/journal/viruses/instructions

Figures
Total number of the manuscript figures: 2.
The figures have appropriate figure legends, although the abbreviation “IDU” could be explained in Figure 1.

Tables
Total number of the manuscript tables: 3.
The tables have appropriate titles and information. However, the following abbreviations should be explained in Tables 1 (“IDU”) and 3 (“OR”, “95% CI”, “aOR”, “a95% CI”).

Author Response

Thank you for the time you have taken to provide a careful and considered review of this manuscript. We have taken on board the feedback you have provided and used it to improve the quality of this manuscript. Please see our responses to your feedback below.

  1. Abstract: We have removed the citations from the abstract and defined all acronyms.
  2. Keywords: We included the MeSH terms “hepatitis C”, “retention in care” and “hospitals”. We also included “micro-elimination”, which is not a MeSH term but is a central theme in this manuscript. We decided against changing “people who use drugs” to “drug users” as, despite not being a MeSH term, it is more consistent with the accepted language used to describe this population.
  3. Abbreviations and acronyms have been explained the first time they are used.
  4. We have described the statistical analysis used in more detail.
  5. All citations have been checked to ensure they are correct.
  6. All references to the supplementary tables throughout the manuscript and in the supplementary materials note have been updated to “Appendix A”, “Appendix B” and “Appendix C”.
  7. The references have been corrected to the appropriate format.
  8. Abbreviations in the figures and tables have been defined.

Reviewer 2 Report

Comments and Suggestions for Authors

This paper addresses the decline in hepatitis C testing and treatment in Australia over the past few years and suggests increased testing in hospitalized patients as a means to reverse this trend. The authors chose two populations - those with diagnoses of injection drug use and those with diagnoses of HCV - to assess retrospectively for evidence of HCV testing. They found that 83% of individuals with these designations were not tested for HCV. 

While the data in the paper would be an important starting point for a local quality improvement project, on its own it is not particularly novel. Even as a simple descriptive analysis of the state of HCV screening in the authors' hospital system, the methods need to be better justified. 

Background 

1. While HCV screening for all persons with history of injection drug use (IDU) makes sense, it's not clear why the authors felt all people with history of HCV should be retested. It is especially unclear why antibody tests would be indicated, and if patients will be referred to outpatient clinics for treatment, it's unclear why they should be retested for HCV while admitted or receiving care in the ED. Did these patients have HCV tests in the electronic medical record? Perhaps adding clarity around the indications for testing people with a history of HCV could better explain why these patients are considered a high priority group for testing. If the goal is to treat people who are hospitalized, do they need recent HCV RNA results to get treatment? For those who are seen in the ED but not hospitalized, what would be the indication for repeating an RNA test? 

There is a comment in the background about targeted testing being preferred to universal testing for hospital-based HCV screening. That statement does not appear to be supported in recent literature, or if it is, more evidence to that effect would be helpful. US screening guidelines have gone from risk-based to universal screening and many hospitals now screen all inpatients. Similarly, studies of risk-based vs non-risk-based screening in EDs are underway and preliminary data indicates that non-targeted screening is superior. (Microsoft PowerPoint - DETECT Hep C Screening Trial CROI Poster 2024 NEW.pptx (croiconference.org))

Methods

In the table of ICD codes, there does not appear to be a diagnosis specific to injection drug use. If the nonspecific drug use codes were used to create the cohort, the indication for testing should be "drug use" rather than IDU. Unfortunately, assuming that all who have the codes for drug use in their files use via injection would be quite inaccurate in imprecise. The codes for HCV are similarly imprecise with the include of "viral hepatitis." Were these cases reviewed through chart review? If so, that step should be described in the methods. 

Results

The results need to be more clearly written. They are hard to follow in many places. For example, line 126 jumped from individuals to episodes without context. The changing denominators are a bit hard to follow and could be simplified or at least better explaned.

The use of the term "inferred" is not defined and it's unclear why 
inferred results are considered valid. 

Very few people who tested positive for HCV antibodies got RNA tests. Reflex RNA testing is standard of care in many places. It would be helpful to include a statement about the lack of reflex testing or any changes to standard practice of the duration of the study period. 

The label on the first column appears to be incorrect in Figure 2. 

In the discussion, more speculation about reasons for testing disparities by department could be included. Also, this would be a good place to review QI and research projects that have successfully increased ED-based and hospital-based testing and treatment. Some suggestions are below. 

Zhou J, Wang FD, Li LQ, Chen EQ. Management of in- and out-of-hospital screening for hepatitis C. Front Public Health. 2023 Jan 25;10:984810. doi: 10.3389/fpubh.2022.984810. PMID: 36761331; PMCID: PMC9905736.

Kela-Murphy N, Moore MS, Verma CM, Bresnahan MP, Harrison E, Schwartz J, Winters A. The Hepatitis C Clinical Exchange Network: A Local Health Department Partnership With Acute Care Hospitals to Promote Screening and Treatment of Hepatitis C Virus Infection. J Public Health Manag Pract. 2022 Mar-Apr 01;28(2):E413-E420. doi: 10.1097/PHH.0000000000001402. PMID: 34347654.

Galbraith JW, Anderson ES, Hsieh YH, Franco RA, Donnelly JP, Rodgers JB, Schechter-Perkins EM, Thompson WW, Nelson NP, Rothman RE, White DAE. High Prevalence of Hepatitis C Infection Among Adult Patients at Four Urban Emergency Departments - Birmingham, Oakland, Baltimore, and Boston, 2015-2017. MMWR Morb Mortal Wkly Rep. 2020 May 15;69(19):569-574. doi: 10.15585/mmwr.mm6919a1. PMID: 32407307; PMCID: PMC7238951.

Williams J, Vickerman P, Smout E, Page EE, Phyu K, Aldersley M, Nebbia G, Douthwaite S, Hunter L, Ruf M, Miners A. Universal testing for hepatitis B and hepatitis C in the emergency department: a cost-effectiveness and budget impact analysis of two urban hospitals in the United Kingdom. Cost Eff Resour Alloc. 2022 Nov 14;20(1):60. doi: 10.1186/s12962-022-00388-7. PMID: 36376920; PMCID: PMC9664679.

Le E, Chee G, Kwan M, Cheung R. Treating the Hardest to Treat: Reframing the Hospital Admission as an Opportunity to Initiate Hepatitis C Treatment. Dig Dis Sci. 2022 Apr;67(4):1244-1251. doi: 10.1007/s10620-021-06941-3. Epub 2021 Mar 26. PMID: 33770327.

Midgard H, Malme KB, Pihl CM, Berg-Pedersen RM, Tanum L, Klundby I, Haug A, Tveter I, Bjørnestad R, Olsen IC, Finbråten AK, Dalgard O. Opportunistic Treatment of Hepatitis C Infection Among Hospitalized People Who Inject Drugs (OPPORTUNI-C): A Stepped Wedge Cluster Randomized Trial. Clin Infect Dis. 2024 Mar 20;78(3):582-590. doi: 10.1093/cid/ciad711. PMID: 37992203; PMCID: PMC10954343.

Author Response

Thank you for the time you have taken to provide a careful and considered review of this manuscript. We have taken on board the feedback you have provided and used it to improve the quality of this manuscript. Please see our responses to your feedback below.

  1. We have added further detail on how codes for hepatitis C are assigned to add clarity to our rational for including them. We are not suggesting that all patients with a history of hepatitis C be re-tested, however due to the nature of how these codes are derived, there will be a subset who have not had adequate testing for a diagnosis. A further subset who, despite previously being diagnosed, require repeat testing to confirm they have a current chronic to be eligible for subsidized treatment.
  2. Thank you for highlighting some key studies we had overlooked. We have revised and expanded the section of the background describing the role of emergency departments and hospitals to better reflect the current literature. This revised section now highlights the importance of a micro-elimination approach for hepatitis C.
  3. Unfortunately, there is not an ICD-10 code for injecting drug use. The codes we used have previously been validated as an indicator of injecting drug use by Curtis et al. 2022. The methods have been updated to justify the use of these codes.
  4. Some revisions have been made to improve the readability of the results section.
  5. We have defined the term “inferred” in the methods (page 3, line 131). These results are derived from measured tests further along the care cascade. For example, if a patient with an HCV RNA test we can infer that they are antibody positive, as the Medical Benefits Scheme does not fund HCV RNA tests without a positive antibody test prior. We felt that the inclusion of inferred results in the care cascade provided a more complete picture of the current state of hepatitis C care at our hospital. We have included a description of the current standard of care and funding structures to provide additional clarity.
  6. Thank you for this suggestion, we have included a statement to this effect.
  7. This figure is correct. The N for the first column is the total number of patients with either a code for IDU or a code for hepatitis C.
  8. Again, thank you for suggesting these studies. We have made considerable revisions to the discussion. We feel these revisions have strengthened the implications of the findings in this study and provided greater insight into the role of hospitals and emergency departments in micro-elimination of hepatitis C. 

Reviewer 3 Report

Comments and Suggestions for Authors

Congratulations to all the authors for this manuscript. I think it adds to the current literature and highlights an important issue - that patients at risk of HCV are not regularly being offered HCV specific care when they enter the hospital system.

Would recommend for publication with some minor suggestions

A few suggestions/comments

1.      I would have liked the authors to have made more of their findings in the discussion – it shows that the business-as-usual approach is not working for hospital in-patients and perhaps a change of paradigm is needed with systemic ways to ensure better rates of HCV testing and LTC

2.      It needs be made clearer that IDU (or past IDU) is the main risk factor for HCV that has been assessed in this study. At times the authors talk about people “at risk of living with HCV” more broadly – in the abstract, in the first sentence of study design and the primary study outcome. However, there are other known non IDU risk factors that have not be included in this study (CALD background from high endemic countries, overseas/unsterile medical procedures etc).

3.      It should be acknowledged that patients may not disclose risk factors for HCV particularly in a hospital setting or with a new clinician. Also, hospital doctors may not routinely ask so patients at risk may have been missed.  

4.      It should be discussed further that you have only assessed 2.3% of all admissions. It is very likely that in the other 97.7% there would have been people with HCV. It is also plausible that the rates of testing/LTC in this group would be even worse.

5.      RE: Data analysis section. For logistic regression have you performed both univariate and multivariate analysis? It was unclear to me from reading this.

6.      Re: comparing number of at risk patients by admission specialty (page 6). A few issues with this.

a.      Do you have the denominator of the total number of admissions for each of those specialties - because medical admissions are generally greater than surgical admissions in any hospital.  Looking at the absolute number is less useful that looking at percentage of total admissions in that specialty

b.      Have these been compared with a statistic test?

7.      Do you have data on country of birth of patients admitted to hospital? If so, how many were overseas born?

8.      It may also be useful to add an additional sentence or two describing the demographics of the hospital more (SES, private insurance status, CALD or first nations populations)(as this information will not be known to international readers and is helpful for when they are trying to compare to their own countries/ populations)

9.      It may be useful to reference the SEARCH project (https://doi.org/10.1111/jvh.13393) Up to the authors. However there are some features that I think justify this and also complement your work

a.      Similar Australian data on ED/hospital HCV testing from a different state/patient population

b.      Also showed rates of hospital testing and LTC very poor prior to intervention (see section 3.5)  

c.      Showed that in up to 1 in 4 hospital patients with HCVAb no risk factor is identified and less than 60% had IDU (either past or current).

Well done to the whole team.

Comments on the Quality of English Language

No significant issues with language identified

Author Response

Thank you for the time you have taken to provide a careful and considered review of this manuscript. We have taken on board the feedback you have provided and used it to improve the quality of this manuscript. Please see our responses to your feedback below.

  1. We have made considerable revisions to the discussion. We feel these revisions have strengthened the implications of the findings in this study and provided greater insight into the role of hospitals and emergency departments in micro-elimination of hepatitis C.
  2. Thank you for this suggestion, we have made edits to the abstract and aims to make it clear that we are only referring to people with a risk due to injecting drug use.
  3. Patient disclosure and how routinely healthcare professionals ask is certainly another challenge to providing hepatitis C care, thank you for highlighting this. We have included a statement to this effect in the limitations section of this study.
  4. We have expanded the role of ED and hospitals discussion. In this we have included that one of the benefits of universal screening is identifying patients with hepatitis C who do not have identifiable risk factors.
  5. It was a univariate logistic regression adjusted for age and sex. The methods have been updated to clarify this.
  6. The denominator for antibody testing is the number of unique patients admitted to that specialty. Only the admissions column and unique patients column use the denominator for the whole subgroup, and this was to show the distribution of admissions and patients across the specialties. We did not compare the results in this table using a statistical test. The factors impacting the odds of hepatitis C care were assessed for each specialty individually in the logistic regression analysis.
  7. Unfortunately cultural and ethnicity data is encouraged to be collected on admission but is not reliably documented. The inclusion of this data would have been very informative if it were reliable.
  8. We have included a description of the demographics of the region, notably the proportion born in non-English speaking countries, the proportion who identify as Aboriginal and Torres Strait Islanders and the estimated prevalence of hepatitis C.
  9. Thank you for recommending this study, it is very relevant and we have included it.

Round 2

Reviewer 2 Report

Comments and Suggestions for Authors

Suggestions were incorporated and the manuscript has been improved.